# Depressive and Anxious Symptoms in Hepatitis C Virus Infected Patients Receiving DAA-Based Therapy

**DOI:** 10.3390/diagnostics11122237

**Published:** 2021-11-29

**Authors:** Claudia Monica Danilescu, Daniela Larisa Sandulescu, Mihail Cristian Pirlog, Costin Teodor Streba, Ion Rogoveanu

**Affiliations:** 1Doctoral School, University of Medicine and Pharmacy Craiova, 200349 Craiova, Romania; monica.danilescu@gmail.com; 2Department of Gastroenterology, Faculty of Medicine, University of Medicine and Pharmacy of Craiova, 200349 Craiova, Romania; larisasandulescu@yahoo.com (D.L.S.); ionirogoveanu@gmail.com (I.R.); 3Department of Medical Sociology, Faculty of Medicine, University of Medicine and Pharmacy of Craiova, 200349 Craiova, Romania; 4Department of Scientific Research Methodology, Faculty of Medicine, University of Medicine and Pharmacy of Craiova, 200349 Craiova, Romania; costinstreba@gmail.com

**Keywords:** Hepatitis C virus, Direct Acting Antivirals, depression, anxiety, fibrosis, sustained virological response

## Abstract

Hepatitis C virus (HCV) represents the most important etiologic factor for advanced fibrosis/cirrhosis and hepatocellular carcinoma associated with a psychological dimension. Our study aims to assess, on a sample comprising of 90 HCV-infected subjects (96.67% F3–F4 METAVIR), the relationship between Direct-Acting Antiviral (DAA) therapies and the psychological effects of the liver disease, focused on the anxious and depressive symptoms. The comprehensive evaluation was done before starting the DAA treatment (BSL), after 12 weeks (End of Treatment—EOT), respectively after another 12 weeks (Sustained Viral Response—SVR). Presumable depressive and/or anxious symptoms were evaluated by Hospital Anxiety and Depression Scale (HADS). The reported depressive symptoms decreased from 21.11% (BSL) to 1.11% (SVR) (*p* < 0.00001), while the anxious ones dropped from 43.34% (BSL) to 4.44% (SVR) (*p* < 0.00001), without a clear evolutionary pattern. We identified no statistically significant interaction between comorbidities (anemia, CKD, obesity) over HADS scores evolution (*p* > 0.05), while the DAAs side-effects (fatigue, headache, pruritus) significantly influenced the anxious and depressive symptoms (*p* < 0.05). During and after the DAA-based therapy, patients with HCV infection presented a significantly reduced rate of the associated depressive and anxious relevant symptoms.

## 1. Introduction

The HCV infection is one of the most frequent causes of liver cirrhosis in the world with consequent significant social and health burdens [1]. Until 2011, pegylated interferon (peg-IFN) and ribavirin (RBV) therapy were used in the treatment of hepatitis C all over the world. The chance of success with this treatment was 30–40%. Success rates have reached 90–95% with the combination of Ombitasvir/Paritaprevir/Ritonavir/Dasabuvir, which is used worldwide since 2014 [2,3,4]. It is known that patients with chronic medical illness, such as liver cirrhosis, may be afflicted by symptoms of depression [5] and anxiety [6], especially in the end stage of their liver disease, which is characterized by frequent medical complications, hospitalization, functional limitation and change of body image. In patients with HCV infection, alterations in health-related quality of life (HRQoL) and neuropsychological disturbances were described also in the absence of liver cirrhosis [7,8]. In fact, even in the absence of debilitating symptoms, HCV may adversely affect the HRQoL by negatively impacting the physical and mental well-being of the patients [9]. Many authors have postulated a direct action of the HCV at the central nervous system level and viral replicates have been detected in the cerebral tissue, but it is difficult to determine whether the neuropsychiatric symptoms are due to direct neurotoxicity of the virus per se or to emotional stress related to the functional deficit, social stigma and apprehension for long-term prognosis [10,11]. In the past, patients with a diagnosed mental health disease (MHD), such as major depression, bipolar disorder, schizophrenia, generalized anxiety, and post-traumatic stress disorder or requiring anti-depressants, antipsychotics, mood stabilizers, or psychotropic drugs prescribed by a psychiatrist, were marginalized with respect to HCV therapy and MHD was one of the most frequently cited reason for exclusion from interferon-based therapy [12]. During the last years, HCV therapy has evolved from interferon-based to Direct Acting Antivirals (DAA), with excellent tolerability and efficacy [13]. Several studies investigated the modifications of HRQoL after HCV eradication [14,15,16].

Our study aims to assess the influence of liver disease, its treatment with DAAs and biological status, toward the psychological status of the subjects, focused on anxiety and depressive disorders.

## 2. Materials and Methods

A prospective and single-centered study was conducted between August 2017 and December 2018 in the Gastroenterology Clinic of the Emergency Hospital Craiova, Dolj County, Romania.

The study sample consisted of 90 consecutive patients diagnosed with HCV infection: compensated hepatic cirrhosis (Child Pugh A or B) or hepatitis. During the study period, all the subjects were under treatment with DAAs: Ombitasvirum/Paritaprevirum/Ritonavirum/Dasabuvirum combination for 12 weeks, according to the Romanian National Protocol for HCV infected patients (2017) [17] and to the guidelines of the Romanian Society of Gastroenterology and Hepatology (RSGH) [18,19] respecting the recommendations of the European Association for the Study of the Liver [4]. These patients were recruited from those who received DAAs treatment in the Romanian national program of interferon-free therapy for HCV infection.

We considered including in the study patients aged 18–80 years, being treatment naïve or previously treated with interferon, without neurologic and psychiatric comorbidities, and without being under any psychiatric medication, for the last 12 months. All these data that referred to the personal medical history of each patient were obtained during the first evaluation visit that they needed to pass through in order to become eligible for the Romanian National Program for HCV interferon-free therapy. Moreover, at the baseline moment of our study, the medical history of the patients was checked again by the gastroenterologist, who rechecked the personal records of the patient and also questioned him/her about psychiatric status during the structured clinical interview, in order to fulfill our study’s inclusion criteria.

The assessment of the biological and psychological status was done at the following three moments of the study (Figure 1):Baseline (before the initiation of the DAA treatment—BSL);End of the DAA treatment (end of treatment—12 weeks after baseline—EOT);Follow-up (Sustain Viral Response—12 weeks after the end of the treatment—SVR).

**Figure 1 diagnostics-11-02237-f001:**
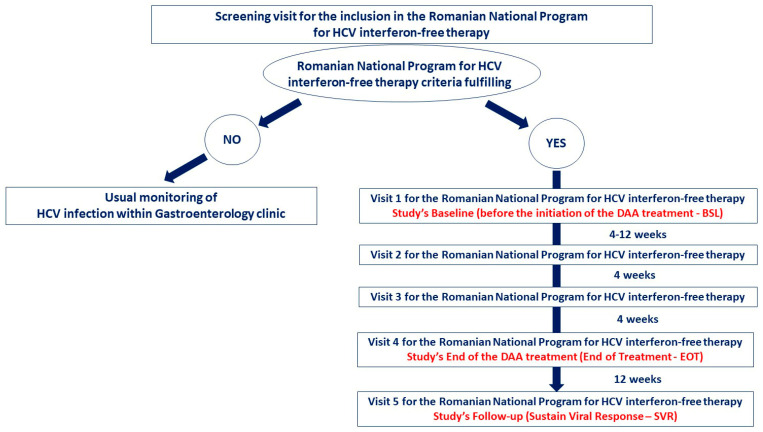
Patients flow diagram for the whole process of DAAs intervention. The study arm consisted of 90 subjects from BSL to SVR.

The subjects were evaluated upon performing a clinical examination and by taking a detailed medical history, including interferon-based treatment history and comorbidities. We also recorded the participants’ demographic (gender, age, residence), clinical (height, weight, Body-Mass-Index—BMI) and biological data (results of Fibrotest, Hepatitis C virus ribonucleic acid viral load–HCV-RNA, Child scores and blood samples).

Fibrosis degrees were classified according to international guidelines by the FibroTest^®^ and/or FibroScan^®^ methods (F0, F1, F2, F3 or F4). HCV-RNA was obtained from blood samples, and the biomedical parameter collected were hemoglobin (mg/dL), alanine aminotransferase (ALT) (IU/L), aspartate aminotransferase (AST) (IU/L), total bilirubin (mg/dL), direct bilirubin (mg/dL), gamma glutamyl transpeptidase (GGT) (IU/L), albumin (g/dL), alpha-fetoprotein (IU/mL), creatinine (mg/dL), and international normalized ratio (INR).

In order to identify and measure the severity of the anxious and depressive symptoms, we used the Hospital Anxiety and Depression Scale (HADS), a self-reported screening tool that includes 14 multiple-choice questions. There are seven depression items measuring cognitive and emotional aspects of depression, predominantly anhedonia, intermingled with seven anxiety items that focus on cognitive and emotional aspects of anxiety. There are anxiety subscale (HADS-A) and depression subscale (HADS-D) both with a four-point Likert scale, ranging from 0–3—score range 0–21 for the HADS-A, respectively HADS-D, while higher scores indicate greater severity. Zigmond and Snaith [20,21] recommend the following cutoff scores for the subscales: 0–7 = non-case, 8–10 = possible case (borderline), and 11–21 = probable case, in our study being used these cutoff scores, which proved for anxiety a specificity of 0.78 and a sensitivity of 0.9, and for depression a specificity of 0.79 and a sensitivity of 0.83 [22]. The HADS questionnaire has been considered useful for the initial diagnosis of depression and anxiety by the National Institute for Health and Care Excellence (NICE) [23] and in our study, we have used the previously validated version for the Romanian population [24].

The study’s questionnaires were explained and administered by the gastroenterologist, during the scheduled consultations for the national program that matched the study’s moments (BSL, EOT, SVR), in a room dedicated to the study’s activities. The filled questionnaires were collected by the research assistant right after the patient completed them and recorded in the secured digital database. The patients identified by HADS results as probable cases for anxiety or depression were sent for counseling and further assessment/monitoring to the psychologist of the Gastroenterology Clinic of the Emergency Hospital Craiova. Based on this supplementary evaluation, those possible individuals considered at risk would be referred to psychiatry.

There were no drop-out patients recorded during the whole research time interval. The facts that contributed to this rate were the following: (1) all subjects were included in the Romanian National Program for HCV interferon-free therapy, which required finishing the treatment unless some serious side-effects or other medical conditions intervened; (2) it was mandatory for all these patients to follow the Romanian National Program’s schedule which included laboratory tests, ultrasound evaluation, and clinical exams at five moments (week 0, week 4, week 8, week 12, week 24), three of these moments overlapping our study’s milestones; (3) all patients were voluntarily selected and, during the inform consent process, they understood the importance of our study and all of them have shown a proactive behavior; (4) before, during, and after the therapy period, a good professional collaboration between the gastroenterologist, the research assistant, and the patient was established, which was an important aspect that contributed to this zero drop-out rate.

The study was approved by the Ethics Committee of Craiova University of Medicine and Pharmacy of Craiova and was in line with the Helsinki Declaration. All patients were enrolled voluntarily upon submission of written informed consent and their data were kept secure.

### Statistical Analysis

The study database included 18 factors as follows: demographic, laboratory investigations, comorbidities, previous HCV treatment, and, for the experienced ones, the quality of response on interferon-based therapy (discontinuation due to side-effects, non-responders, relapse), and the assessment of the psychological status.

Descriptive analysis for continuous variables was performed with mean ± standard deviation for normally distributed data and the qualitative variables as absolute and relative frequencies (%).

We used the Friedman test, a non-parametric test for repeated measures, to test for differences between groups when the dependent variable being measured was ordinal. The categorical data were compared with the Chi-square test (χ^2^). Comorbidities’ interaction on the evolution of HADS scores over time was tested using the two-way mixed ANOVA statistical test. We used the Shapiro–Wilk’s test for data normality analysis, Levene’s test of homogeneity of variances and Box’s M test for variances and covariances, as well as Mauchly’s test for sphericity. The following *p* values were accepted: *p* < 0.05 significant in a confidence interval (CI) of 95%, *p* < 0.01 (CI of 99%), *p* < 0.001 highly significant (CI of 99.9%). Using Kendall’s W as an effect size measurement for Friedman’s test, and choosing 0.2 as the effect size value, for a significance level α = 0.05 and a power 1 − β = 0.8, for three groups of repeated measurements, we have suggested a sample size of 42 persons (G*Power 3.1.9.7, Heinrich Heine University Düsseldorf, Düsseldorf, Germany). Even with the 15% correction of the sample size, we would need 49 subjects as the minimum sample size for the desired levels for effect size, significance level and power, for the described study design.

All statistical analyses were performed by IBM SPSS Statistics 25.0 (Chicago, IL, USA), while the primary data were recorded in Microsoft Excel files.

## 3. Results

The present study comprises data from 90 HCV infected patients that were under treatment with DAA for the 12 weeks after the recruitment (76.63% women; mean age 67.71 ± 7.83 years, age ranging from 33 to 79 years, 53.3% of them living in an urban environment) (Table 1).

At BSL, 53 patients (58.88%) were classified as affected by well compensated cirrhosis (48 Child-Pugh class A and five Child-Pugh class B) on clinical and radiological findings. The degree of fibrosis according to the METAVIR score was F2 in three patients (3.3%), F3 in 34 (37.77%), respectively F4 in 53 (58.88%) (Table 2).

The mean viral load was 1,178,736 ± 1,387,127 IU/mL (BSL), decreased to zero (EOT) and kept the same zero value (SVR) for all the study subjects. We identified 64 patients (71.11%) as therapy-naïve, while the remaining 26 (28.88%) had been treated before with combinations of pegylated interferon and ribavirin. 

Fifteen subjects (16.67%) were identified with diabetes mellitus and 26 (28.89%) fulfilled the criteria for obesity class I and II (BMI > 30 kg/m^2^). Chronic kidney disease (CKD) was present on G2 stage in 58 (64.44%) of the subjects, respectively G3 stage on four patients (4.44%), while anemia (hemoglobin level less than 13.5 mg/dL in males and less than 12 mg/dL in female) was found in 22 subjects (26.67%) (Table 2). For all identified comorbidities found at BSL, which were under specific treatment, their current therapeutically status was kept. During the all-study period, those patients were monitored both by the gastroenterologist and the specialist physician, who was on duty for the comorbidity’s therapy. While the treatment of comorbidities was kept as previously established before the start of the HCV therapy, we observed that the improvement of the psychological status appeared after the DAAs treatment. Thus, we could consider that the changes in mental health were directly related to HCV therapy.

Before starting DAA therapy, the level of Hemoglobin was 13.26 ± 1.73 mg/dL, the levels of transaminase, as we expect, were higher (ALT 105.25 ± 86.99 IU/L, AST 86.11 ± 55.79 IU/L) as well as GGT (68.13 ± 37.79 IU/L) and Alpha-fetoprotein (11.05 ± 16.45 IU/L). Total bilirubin (1.02 ± 0.59 mg/dL), albumin (4.06 ± 0.5 g/dL), INR (1.05 ± 0.21) and creatinine (0.77 ± 0.14 mg/dL) mean levels were between normal limits at baseline (Table 3).

The evaluation of the psychological status in all three moments did not show a clear pattern of the severity of the anxious and depressive symptoms. Thus, at BSL, 39 probable cases of anxiety (43.33%), respectively 19 (21.11%) probable cases of depression, out of which seven individuals (7.78%) presented associated anxious and depressive significant symptoms. During the study period, it was revealed that HADS scores decreased for both psychiatric disorders, which meant that the patients’ mental health had improved in a significant way (Table 4).

Interestingly, at SVR, the scores of the HADS showed that depressive symptoms were clearly improved, while anxious ones remained in the probable case category in only four subjects (4.44%) (Figure 2, Table 4). The dynamic measures of HADS scores showed a highly significant difference both for anxiety scores (Friedman test χ^2^ = 53.19, *p* < 0.00001), and for depression (Friedman test χ^2^ = 26.23, *p* < 0.00001). This fact could be considered a relevant indicator of the improvement of the mental health status during the DAA treatment.

Analyzing the psychological impact of the HCV infection and its treatment according to the gender of the patients, we found that 47.82% of women had clinically relevant anxious symptoms before DAAs initiation (HADS-A 13.21 ± 1.96) to an EOT score of 7.18 ± 2.96, while at the SVR the same score decreases to 4.72 ± 3.06 (Friedman test χ^2^ = 44.65, *p* < 0.00001), similarly to the male patients (BSL 13 ± 1.78; EOT 6.83 ± 2.78; SVR 4 ± 1.89; Friedman test χ^2^ = 9.33, *p* = 0.0004). The same phenomenon was recorded for the depressive symptoms identified in our subjects based on HADS-D scores, especially for the female subgroup (BSL 12.11 ± 1.69; EOT 3.35 ± 3.18; SVR 3.17 ± 2.78) (Friedman test χ^2^ = 22.85, *p* = 0.00001) (Table 4).

Moreover, analyzing the evolution of the average HADS-A scores in normal patients, we identified an increasing trend in the level of severity of anxious symptoms, from BSL 4.9 ± 1.91 to the EOT 8.55 ± 2.74 (borderline cases of anxiety), and decreasing to normality at SVR 5.75 ± 3.55 (Friedman test χ^2^ = 9.175, *p* = 0.01018) (Table 4). This process was also more prevalent in women, where the mean score starts at BSL 4.85 ± 1.91, increase to EOT 8.85 ± 2.65, and ended at SVR 5 ± 3.5 (Friedman test χ^2^ = 10.71, *p* = 0.0047).

There existed a certain number of individuals who reported increased scores on HADS-A (23 subjects—25.5%), respectively HADS-D (19 subjects—21.1%) at SVR, compared to EOT. To analyze these groups of patients we extracted what we considered important components that could have an influence on the evolution of the HADS score from the clinical and psychological points of view (Table 5).

We analyzed anemia, CKD and obesity as comorbidities with potential influence on the HADS scores’ evolution, for both anxiety and depression. For each subgroup, there were very few outliers, as assessed by boxplot, but they were maintained in our analysis, as the study lot was considered large enough. HADS values were normally distributed, except for very few groups. There was homogeneity of both variances and covariances, and Mauchly’s test of sphericity suggested that the assumption of sphericity was indeed met for the two-way interaction between comorbidities’ presence and scores values. We identified no statistically significant interaction between the presence of anemia, CKD and obesity over HADS scores evolution (Table 6).

The periodical clinical assessment imposed by the Romanian National Program for HCV interferon-free therapy (Figure 1) has revealed the presence of DAAs side-effects during the treatment administration, but for our study’s purposes, we have evaluated them at EOT, respectively SVR. At EOT, these adverse effects were recorded in almost a third of the whole study sample, and, similar to the literature [2,12,15] and clinical guidelines [4,18,19] there was pruritus (12.22%), fatigue (8.88%), headache (6.66%) and insomnia (6.66%). Twelve weeks later, at SVR, the medication’s side effects were not present (Table 7).

Analyzing the impact of adverse phenomena on these subjects’ mental health, we found out statistically significant differences (Friedman test) regarding the correlation of fatigue, headache and pruritus with anxious and depressive symptoms, as expressed by the HADS scores. This finding was not applicable for insomnia (Table 7).

## 4. Discussion

The clinical picture of the patients with HCV infection includes numerous symptoms, such as joint pain, itching, loss of appetite, weakened libido, protein energy malnutrition [25,26,27] jaundice, increased bleeding, abdominal distension, ascites, peripheral edema, dyspnea, weight loss, muscle cramps, and the inability to work that are leading to mental health disruption [28,29]. Moreover, it was well known that the peg-IFN and RBV treatment represents a cause for major depression in around 30% of patients [30] and many other side-effects, such as medical manifestations (fatigue and weakness) [2,4,12], or neuropsychiatric symptoms (cognitive deficit, sleep disorders, anger/hostility, anxiety) [31,32] and, in these conditions, adjunctive therapy (e.g., pharmacological, psychotherapy or psychological counseling) must be considered in order to control those side effects.

Among these outcomes could be some other issues related to the association between liver infection and psychopathology, such as mild cognitive deficits, stigma and lower acceptance of illness, history of illicit drug use, lower work capacity, or decreased social support [33,34]. This plethora of side effects was proven to be one of the most important reasons for the treatment discontinuation [35].

Since 2011, DAAs were introduced as a therapy for HCV. They were proven to have optimal safety profiles, higher efficacy, shorter duration of treatment, increased barriers to resistance [36,37,38] and, most importantly in our context, do not favor the onset of psychiatric disorders, while being well-tolerated [39,40]. Additionally, previous studies have shown that the DAAs lead to a significant improvement in the results of the neuropsychological tests after HCV eradication [41,42,43]. 

As mentioned above, for our study sample a clear evolutionary pattern related to the patient-reported anxious and depressive symptoms could not be underlined. The process accomplished through the mental health status evaluation in all three moments have shown an important incidence of both psychiatric symptoms at the starting point of the study, with a non-linear evolution during the therapy period (Figure 2), a phenomenon that was not revealed by previous studies [41,42,44].

A particular situation within our sample was the one linked to those seven subjects identified with both categories of psychological symptoms, in which case we have noticed that the positive outcomes of the infection’s therapy were accompanied by a complete remission of the mood disturbances. The positive influence of the DAAs on the mental status of these subjects was even better than the one presented in other studies [45].

The results of our study showed a direct association between the biological status of the HCV infection (METAVIR scores and laboratory data) and the impairment of the mental health, the worsening of one aspect conducting to a similar effect on the other (Table 5). Thus, based on these observations, the importance of the DAAs intervention could be affirmed as early as possible once the HCV infection was identified and diagnosed, in order to avoid further complications.

Since data from literature [13,15] have shown that comorbid conditions associated to the HCV infection have an impact on the mental health of the affected individuals, another objective of the research was to analyze the characteristics of the study sample from this point of view too. The medical conditions diagnosed in our patients also had an influence on their psychological status, in this context being recommended to assess the overall patients’ status in order to detect the possible comorbidities and to keep them under control to reduce their influence on the HCV infection’s evolution and mental health.

We consider that one of the limitations of this study was that the subjects’ mental status was assessed only by guided self-administrated questionnaires without a structured clinical interview done by a qualified psychiatrist/psychologist. Conversely, a strength of the research is represented by the use of a widely-known and friendly screening instrument for anxiety and depression, and prospective assessments every 12 weeks in terms of clinical, biochemical, virological and psychological monitoring. Moreover, our study was conducted on a well-characterized sample, which reflects real world patients, thus increasing the applicability of the findings to a similar population.

## 5. Conclusions

DAAs therapy proved to be efficient for remitting HCV infection and improving the mental health status: HCV-RNA decreased from a mean value of 1,178,736 IU/mL to zero, while identified depressive and anxious symptoms were significantly enhanced, even if a clear evolutionary pattern could not be highlighted. The psychological disturbances were correlated with the medication side effects, but there could not be found a significant influence by the HCV associated comorbidities. Thus, our data sustain the positive effects of the new HCV therapy for achieving both the remission of the liver disease and the reduction of depressive and anxious symptom severity.

## Figures and Tables

**Figure 2 diagnostics-11-02237-f002:**
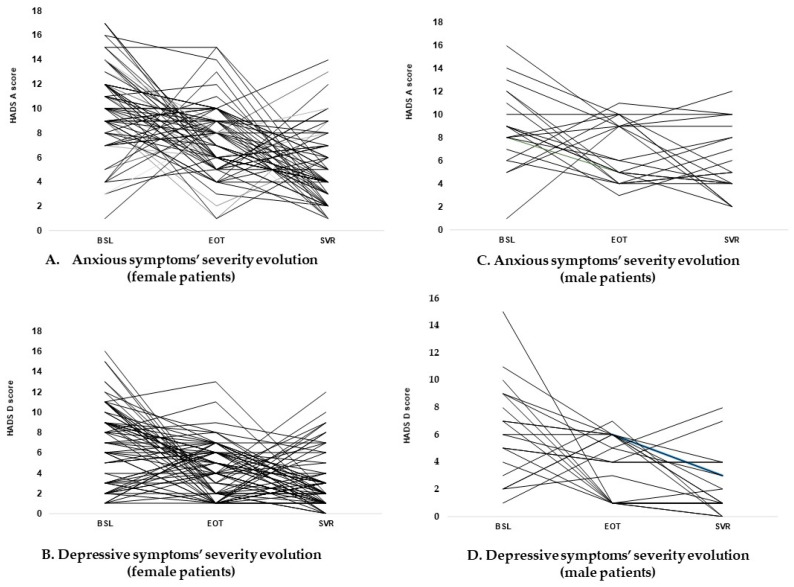
Graphic description of the anxious and depressive symptoms evolution during the study period (gender distribution). HADS scores showed similar patterns of the identified mental health disorders according to the patient’s gender. The dynamic measures of HADS scores showed a significant positive evolution of the psychological status for the subjects included in the study. (**A**) Evolution of HADS-A scores at the three moments of assessment in female subjects; (**B**) Evolution of HADS-D scores at the three moments of assessment in female subjects; (**C**) Evolution of HADS-A scores at the three moments of assessment in male subjects; (**D**) Evolution of HADS-D scores at the three moments of assessment in male subjects.

**Table 1 diagnostics-11-02237-t001:** Demographic data of the study sample.

		(Mean Value ± SD)
Mean age (year)		67.71 ± 7.83
HCV—RNA (UI/mL)		1,178,736 ± 1,387,127
		***n* (%)**
Gender	male	21 (23.37)
	female	69 (76.63)
Environment	urban	48 (53.3)
	rural	42 (46.5)

**Table 2 diagnostics-11-02237-t002:** Patients’ clinical status (BSL).

		*n* (%)
Fibrosis acording to METAVIR	F2	3 (3.33)
	F3	34 (37.78)
	F4	53 (58.89)
Cirrhosis	non-cirrhosis	37 (41.11)
	Child A	48 (53.33)
	Child B	5 (5.56)
Previous HCV Treatment	Treatment naïve	64 (71.11)
	IFN + RBV	26 (28.89)
Interferon base therapy response	discontinuation (side effects)	3 (3.33)
	non-responder	18 (20.00)
	relapse	5 (5.56)
Diabetes mellitus	non insulin-dependent	3 (3.33)
	insulin-dependent	12 (13.33)
BMI (kg/m^2^)	mildly underweight	1 (1.11)
	normal weight	22 (24.44)
	overweight	41 (45.56)
	obese class I	21 (23.33)
	obese class II	5 (5.56)

**Table 3 diagnostics-11-02237-t003:** Patients’ biochemical status (BSL).

	(Mean Value ± SD)	Range
Hemoglobin (mg/dL)	13.26 ± 1.73	8.99–16.00
ALT (IU/L)	105.25 ± 86.99	20–354
AST (IU/L)	86.11 ± 55.79	22–321
Total Bilirubin (mg/dL)	1.02 ± 0.59	0.2–2.8
GGT (IU/L)	68.13 ± 37.79	21–186
Albumin (g/dL)	4.06 ± 0.5	3.1–5.1
INR	1.05 ± 0.21	0.7–2
α-fetoprotein (IU/mL)	11.05 ± 16.45	0.4–87
Creatinine (mg/dL)	0.77 ± 0.14	0.5–1.3

**Table 4 diagnostics-11-02237-t004:** Patients’ psychological status during the study period (χ^2^ Friedman test). On the *p*-Value column, bold is used in order to underline the statistically significant values of *p*.

	*n* (%)	HADS Avg Score BSL	HADS Avg Score EOT	HADS Avg Score SVR	χ^2^	*p*-Value
**All patients**	90 (100)					
HADS-A Probable case	39 (43.33)	13.17 ± 1.87	7.12 ± 2.85	4.61 ± 2.90	53.19	**<0.00001**
HADS-A Borderline	31 (34.44)	9.03 ± 0.76	6.70 ± 2.74	5.61 ± 2,87	20.53	**0.00003**
HADS-A Non-case	20 (22.22)	4.9 ± 1.91	8.55 ± 2.74	5.75 ± 3.55	9.175	**0.01018**
HADS-D Probable case	19 (21.11)	12.21 ± 1.75	3.36 ± 3.11	2.94 ± 2.73	26.23	**<0.00001**
HADS-D Borderline	23 (25.56)	8.91 ± 0.66	4.86 ± 2.76	2.82 ± 2.7	29.76	**<0.00001**
HADS-D Non-case	48 (53.33)	3.93 ± 2.07	4.37 ± 2.18	3.18 ± 2.74	6.385	**0.04106**
**Female**	69 (76.67)					
HADS-A Probable case	33 (36.67)	13.21 ± 1.96	7.18 ± 2.96	4.72 ± 3.06	44.65	**<0.00001**
HADS-A Borderline	22 (24.44)	9.18 ± 0.73	6.77 ± 2.89	5.45 ± 2.8	18.09	**0.00012**
HADS-A Non-case	14 (15.56)	4.85 ± 1.91	8.85 ± 2.65	5 ± 3.5	10.71	**0.0047**
HADS-D Probable case	17 (18.89)	12.11 ± 1.69	3.35 ± 3.18	3.17 ± 2.78	22.85	**0.00001**
HADS-D Borderline	18 (20.00)	8.8 ± 0.67	5.44 ± 2.61	3.16 ± 2.95	22.75	**0.00001**
HADS-D Non-case	34 (37.78)	3.75 ± 2.06	4.41 ± 2.25	3.41 ± 2.82	3.35	0.18703
**Male**	21 (23.33)					
HADS-A Probable case	6 (6.67)	13 ± 1.78	6.83 ± 2.78	4 ± 1.89	9.33	**0.0004**
HADS-A Borderline	9 (10.00)	8.66 ± 0.7	6.55 ± 2.5	6 ± 3.16	3.38	0.1837
HADS-A Non-case	6 (6.67)	5 ± 2.09	7.83 ± 3.06	7.5 ± 3.27	0.58	0.74702
HADS-D Probable case	2 (2.22)	13	3.5	1	-	-
HADS-D Borderline	5 (5.56)	9 ± 0.7	2.8 ± 2.48	1.6 ± 0.89	7.6	**0.02237**
HADS-D Non-case	14 (15.56)	4.42 ± 2.18	4.28 ± 2.08	2.64 ± 2.56	5.82	0.0544

**Table 5 diagnostics-11-02237-t005:** Differences between patients’ groups who had rebound HADS scores at SVR compared to EOT.

	HADS-A EOT < HADS-A SVR Group	HADS-A EOT > HADS-A SVR Group	HADS-D EOT < HADS-D SVR Group	HADS-D EOT > HADS-D SVR Group
*n*	23 (100.00%)	67 (100.00%)	19 (100.00%)	71 (100.00%)
Rural	7 (30.43%)	41 (61.20%)	8 (42.10%)	40 (56.34%)
Urban	16 (69.57%)	26 (38.80%)	11 (57.89%)	31 (43.66%)
Female	16 (69.57%)	53 (79.10%)	17 (89.47%)	52 (73.24%)
Male	7 (30.43%)	14 (20.90%)	2 (10.53%)	19 (26.76%)
Age	63.35 ± 7.96	63.84 ± 7.91	63.57 ± 8.53	62.53 ± 7.76
BMI	27.44 ± 4.42	27.63 ± 4.18	28.49 ± 4.73	27.33 ± 4.07
Previous PegIFN therapy	8 (34.78%)	18 (26.87%)	5 (26.32%)	21 (29.58%)
Diabetes	7 (30.43%)	8 (11.94%)	5 (26.32%)	10 (14.08%)
CKD stage G3 and over	1 (4.35%)	3 (4.48%)	1 (5.26%)	3 (4.23%)
SOT HADS-A	9.70 ± 3.21	9.99 ± 3.76	9.79 ± 1.41	9.94 ± 2.55
EOT HADS-A	4.87 ± 2.26	8.13 ± 2.55	6.68 ± 3.42	7.46 ± 2.62
SVR HADS-A	8.39 ± 2.54	4.12 ± 2.40	7.63 ± 2.85	4.56 ± 1.68
SOT HADS-D	6.48 ± 4.45	7.12 ± 3.64	7.00 ± 4.04	6.94 ± 3.53
EOT HADS-D	3.43 ± 2.25	4.58 ± 2.64	2.26 ± 4.81	4.83 ± 3.59
SVR HADS-D	4.13 ± 3.70	2.67 ± 2.19	6.63 ± 2.14	2.08 ± 3.00

**Table 6 diagnostics-11-02237-t006:** Two-way mixed ANOVA analysis for comorbidities interaction with HADS scores’ evolution over time.

		F (2176)	*p* *	Partial η^2^
HADS-A	Anemia	0.263	0.769	0.003
	CKD	2.488	0.086	0.028
	Obesity	0.629	0.534	0.007
HADS-D	Anemia	0.069	0.933	0.001
	CKD	0.636	0.531	0.007
	Obesity	1.344	0.264	0.015

* Two-way mixed ANOVA.

**Table 7 diagnostics-11-02237-t007:** Side-effects distribution according to the psychological status (χ^2^ Friedman test). On the *p*-Value column, bold is used in order to underline the statistically significant values of *p*.

	Side Effects at EOT	*n* (%)	BSL (Mean ± SD)	EOT (Mean ± SD)	SVR (Mean ± SD)	χ^2^	*p*
**HADS-A scores**	fatigue	8 (8.88)	10 ± 3.20	4.75 ± 2.71	4.75 ± 2.67	8.06	**0.017**
	headache	6 (6.66)	12 ± 2.60	8.16 ± 2.56	4.16 ± 2.04	8.33	**0.0155**
	insomnia	6 (6.66)	9.33 ± 3.38	6.33 ± 1.86	5.83 ± 3	2.33	0.31
	pruritus	11 (12.22)	9.63 ± 3.5	6.09 ± 2.38	4.72 ± 2.41	7.09	**0.028**
**HADS-D scores**	fatigue	8 (8.88)	9.125 ± 4.54	3.5 ± 2.32	1 ± 0.75	9.75	**0.0076**
	headache	6 (6.66)	7 ± 4.14	4.5 ± 1.37	2 ± 2	9.33	**0.0094**
	insomnia	6 (6.66)	4.83 ± 3.98	2.5 ± 2.07	3.5 ± 2.66	1.58	0.45
	pruritus	11 (12.22)	7.63 ± 2.87	4.9 ± 1.64	1.36 ± 0.92	18.13	**0.00013**

## Data Availability

The data used in this study could be available by request and after the approval of the local IRB.

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
