# Peer review of "Depressive and Anxious Symptoms in Hepatitis C Virus Infected Patients Receiving DAA-Based Therapy"

_diagnostics, 2021, doi:10.3390/diagnostics11122237_

Round 1

Reviewer 1 Report

In the current study by Danilescu and et al. the authors conducted a study on HCV positive patients and determined presence of depression and anxiety in these patients and if it improved with HCV treatment. The authors present an interesting article and I suggest the following revisions-

  • The authors fail to note in the methodology if any of these patients had h/o of prior depression or anxiety or were taking any antidepressant meds at home. They need to elaborate on this and how was this history verified.
  • One major critique is that the authors have only provided baseline information about CKD and DM, but many of these patients could be having other comorbidities and thereby contributing to overall anxiety and depression. Unless this is know if patients were baseline healthy or having multiple comorbidities it cannot be extrapolated that anxiety/depression was just related to HCC and treatment with DAA’s improved it.
  • How many patients did not complete the survey at subsequent visits? How many were returned incomplete and how was this missing data calculated? Also, who administered the survey and when was it collected- was it right after the visit or there was a time lag when the repeat surveys were administered. . Too much critical information is missing.
  • Doing subgroup analysis based on Metavir staging, anemia, CKD labs etc is not suggested unless the p value of interaction is significant. Doing these multiple analysis will result in type 1 error. Suggest the authors check for p value on interaction and if not significant removing these results.
  • Ideally it would have been worthwhile to report the percentage of patient reporting symptoms like fatigue, loss of appetite, sleep etc and presenting the HADs scores.

Author Response

Response to Reviewer 1 Comments

Point 1. The authors fail to note in the methodology if any of these patients had h/o of prior depression or anxiety or were taking any antidepressant meds at home. They need to elaborate on this and how was this history verified.

Response 1. We have addressed this comment in the revised version of the manuscript at rows 80-88 (please see the attached manuscript).

Point 2. One major critique is that the authors have only provided baseline information about CKD and DM, but many of these patients could be having other comorbidities and thereby contributing to overall anxiety and depression. Unless this is know if patients were baseline healthy or having multiple comorbidities it cannot be extrapolated that anxiety/depression was just related to HCC and treatment with DAA’s improved it.

Response 2. We have addressed this comment in the revised version of the manuscript at rows 189-200 (please see the attached manuscript).

Point 3. How many patients did not complete the survey at subsequent visits? How many were returned incomplete and how was this missing data calculated? Also, who administered the survey and when was it collected- was it right after the visit or there was a time lag when the repeat surveys were administered. . Too much critical information is missing.

Response 3. We have addressed this comment in the revised version of the manuscript at rows 125-129, respectively 134-145 (please see the attached manuscript).

Point 4. Doing subgroup analysis based on Metavir staging, anemia, CKD labs etc is not suggested unless the p value of interaction is significant. Doing these multiple analysis will result in type 1 error. Suggest the authors check for p value on interaction and if not significant removing these results.

Response 4. We have addressed this comment in the revised version of the manuscript at rows 159-163, respectively 256-268 (please see the attached manuscript).

Point 5. Ideally it would have been worthwhile to report the percentage of patient reporting symptoms like fatigue, loss of appetite, sleep etc and presenting the HADs scores.

Response 5. We have addressed this comment in the revised version of the manuscript at rows 270-283 (please see the attached manuscript).

Reviewer 2 Report

This prospective study analyzed anxiety and depression during and after DAA treatment for HCV eradication.

Comments

  1. The study excluded patients with psychiatric disorder but analyze the trend of change of anxiety and depression by self-reported scores of assessment scales. But "abnormal" scores were noted and treated as "depression" or "anxiety". Can these patients be categorized as "depression" or "anxiety" by only scales? The title may be misleading.
  2. The tend of change in Figure 1 seems to consist of a large portion of patients who had rebound scores on the time points of SVR. Authors should analyze this group of patients and compare their significance.  Besides, what components are involved in the trend of change?
  3. Does the psychological assessment belong to extrahepatic manifestation of HCV disease?
  4. Do all patients finished the three assessments? Any patient excluded for analysis because of not completing all the assessments? What is the patient flow diagram?
  5. Did patients receive any interventions once they were found to have depression and anxiety?

Author Response

Response to Reviewer 2 Comments

Point 1. The study excluded patients with psychiatric disorder but analyze the trend of change of anxiety and depression by self-reported scores of assessment scales. But "abnormal" scores were noted and treated as "depression" or "anxiety". Can these patients be categorized as "depression" or "anxiety" by only scales? The title may be misleading.

Response 1. We have addressed this comment in the revised version of the manuscript at rows 2-3, respectively 110-124 (please see the attached manuscript). Also, in order to solve the above-mentioned issue, in the revised version of the manuscript, the “abnormal” scores were noted as “anxious and/or depressive symptoms”.

Point 2. The tend of change in Figure 1 seems to consist of a large portion of patients who had rebound scores on the time points of SVR. Authors should analyze this group of patients and compare their significance.  Besides, what components are involved in the trend of change?

Response 2. We have addressed this comment in the revised version of the manuscript at rows 247-254 (please see the attached manuscript).

Point 3. Does the psychological assessment belong to extrahepatic manifestation of HCV disease?

Response 3. We have addressed this comment in the revised version of the manuscript at rows 193-200 (please see the attached manuscript).

Point 4. Do all patients finished the three assessments? Any patient excluded for analysis because of not completing all the assessments? What is the patient flow diagram?

Response 4. We have addressed this comment in the revised version of the manuscript at rows 89-97, respectively 134-145, and Figure 1 (please see the attached manuscript).

Point 5. Did patients receive any interventions once they were found to have depression and anxiety?

Response 5. We have addressed this comment in the revised version of the manuscript at rows 129-133 (please see the attached manuscript).

Round 2

Reviewer 1 Report

The authors have answered all the queries. 

Author Response

We would like to thank for the interesting and constructive comments. We have done our best addressing all points raised, and hopefully brought the manuscript to the required higher standard.

Please see the attached manuscript's last version.

Reviewer 2 Report

The authors should revise content of abstract and conclusion according to the revised version.

Author Response

(The authors gave the same response as above.)
